# O₂ and CO₂ Responses of the Synaptic Period to Under-Ice Phytoplankton Bloom in the Eutrophic Razdolnaya River Estuary of Amur Bay, the Sea of Japan

**Pavel Semkin** * , **Pavel Tishchenko** *, **Galina Pavlova, Yuri Barabanshchikov, Petr Tishchenko** , **Maria Shvetsova, Elena Shkirnikova and Yulia Fedorets**

Il'ichev Pacific Oceanological Institute, Far Eastern Branch, Russian Academy of Sciences, 690041 Vladivostok, Russia
* Correspondence: pahno@list.ru (P.S.); tpavel@poi.dvo.ru (P.T.); Tel.: +7-9146647833 (P.S.)

**Abstract:** Hydrological conditions are an important factor for aquatic ecosystems. Their contribution to stimulating phytoplankton bloom in eutrophic estuaries is not quite clear. We present the results of an outbreak of a phytoplankton bloom event observed in the eutrophic Razdolnaya R. estuary in 2022 from January 22 to February 23, when the estuary was covered by ice. The bloom spreads over 21 km from the river mouth bar to upstream in the near-bottom layer below the halocline. The Chl-a concentration in the bloom area increased from 15 to 100 μg/L, and the dissolved oxygen concentration from 350 to 567 μmol/kg at a rate of 11 μmol/(kg day) over the study period, while the $CO_2$ partial pressure was reduced to 108 μatm in the most oxygen-supersaturated waters. The *Thalassiosira nordenskioeldii* Cleve sea diatom was the dominant phytoplankton species in the bloom area. The opposite trend was observed near the boundary of the saline water wedge penetration over 29 km from the river mouth bar to upstream where the dissolved oxygen concentration decreased from 140 to 53 μmol/kg over a month, and partial pressure of $CO_2$ reached 4454 μatm. We also present the results obtained in February 2016 before and after a snowfall, when the ability of PAR to penetrate through the ice was impeded by a layer of snow. After the snowfall, photosynthesis in the under-ice water stopped and the oxygen concentration decreased to almost zero due to the microbiological destruction of the phytoplankton biomass. As such, the main effect of phytoplankton bloom is the formation of superoxia/hypoxia (depending on the light conditions), during the period of maximum ice thickness and minimum river discharge. Thus, this study demonstrates that the eutrophication in the future could lead to unstable ecosystems and large synoptic variations of dissolved oxygen and $CO_2$ partial pressure of the estuaries.

**Keywords:** salt-wedge estuary; phytoplankton blooms; dissolved oxygen supersaturation; hypoxia; partial pressure $CO_2$; ice-covered period; eutrophication



## 1. Introduction

The dynamic phytoplankton bloom in estuaries is very important for aquatic ecosystems since a significant increase in phytoplankton biomass occurs in relatively short periods [1]. The high intensity of photosynthesis during blooming lead to atmospheric $CO_2$ absorption in estuaries [2]. Alternatively, in eutrophic estuaries, there is an issue with rapid oxygen content reduction and water acidification in the bottom layers where light intensity is limited due to high turbidity, and the rate of phytoplankton biomass microbial destruction exceeds that of primary production [3–5]. Estuary basins are very vulnerable due to increasing eutrophication, which is attested by the increase in hypoxia in near-bottom waters [6–10].

A bloom occurs when phytoplankton growth rates exceed losses such that a sustained period of growth leads to the accumulation of biomass. It is known that the key factors

for phytoplankton bloom are an abundance of nutrients and favorable conditions: sufficient photosynthetically active radiation (PAR), suitable salinity and water temperature, and the vertical stability of the water layer [11]. While the decrease in biomass may be associated with natural death, water advection, zooplankton grazing [12,13], and viral lysis [14], blooms are most often associated with the massive growth of diatom, followed by dinoflagellates [15]. The dominance of diatoms is due to their species [16] and their ability to grow rapidly in the nutrient-rich environments that are so common in eutrophic estuaries [17]. The factors affecting the period and intensity of under-ice phytoplankton blooms have been widely discussed for freshwater [18,19] and the marine ecosystems of subarctic and arctic regions [20–22], and Antarctic regions [23].

Many studies have been conducted to understand the value and seasonal pattern of phytoplankton production in partially mixed and vertically homogeneous estuaries [1,17,24–27]. The formation of superoxia/hypoxia has recently been considered in relation to under-ice blooms and water dynamics in seasonally ice-covered eutrophic estuaries [28]. There are reports of phytoplankton blooms in the highly stratified waters of estuaries [29–31]. In such estuaries, density stability prevails over tidal mixing, resulting in a salt wedge with a clear boundary between fresh and seawater [32]. In this case, two types of phytoplankton blooms can be distinguished [33]: type (1), which undergo massive growth in the surface layer and prefer brackish waters; and type (2), which undergo massive growth under the halocline under sufficient light. However, the effects of phytoplankton blooms on the ecosystems of ice-covered stratified estuaries are not clearly understood. There is a need to understand the biochemical processes in such basins as more and more water basins are subjected to eutrophication [34].

The Razdolnaya R. estuary is very eutrophic, and near-bottom hypoxia can be observed in it during both low and high-water periods in summer [10,35]. The dissolved nitrogen and phosphorus concentrations in the ice formation period are up to 500 and 9 µmol/L, respectively [36]. In Amur Bay, the receiving basin of the Razdolnaya R., the phytoplankton blooms in every season, while the phytoplankton species composition usually varies [37,38]. Under the ice, the *Thalassiosira nordenskioeldii* diatoms often dominate in terms of biomass, making up anywhere from 60 to 87% of the total phytoplankton biomass [39,40]. A correlation was found between the Chl-a concentration in the under-ice water and the presence of snow on the ice and PAR was found in the Razdolnaya R. estuary [41].

This study's objective is to identify the phytoplankton bloom outbreak in the synaptic period and the response of characteristics reflecting the organic matter (OM) production/destruction balance in the eutrophic estuary during winter low-water periods with ice formation.

## 2. Materials and Methods

### 2.1. Study Area

The transboundary Razdolnaya R. (the China-Primorye Region in the southeastern part of Russia) flows into the northern part of Amur Bay (Peter the Great Bay, the Sea of Japan) (Figure 1). The estuary measures about 50 km in length and is located within the boggy Razdolnenskaya depression and the northern part of Amur Bay (Figure 1). The Razdolnaya R. catchment area is 16,800 km$^2$. The average discharge of the river for the last 11 years has been 103.5 m$^3$/s (http://gmvo.skniivh.ru/ (accessed on 18 May 2022)). The water regime of Razdolnaya R. is characterized by steady low winter discharge (7.2 and 5.7 m$^3$/s in January (Jan) and February (Feb), respectively) and an absolute minimum discharge of 1.5 m$^3$/s in Feb. Spring floods can be observed in May. The peaks of spring floods are about ten times higher than the average annual discharge of the river. The absolute discharge peaks exceed 3000 m$^3$/s during the summer and fall floods in occasional years. Over the course of the freeze-up period from late November to early April, a saltwater wedge tends to penetrate the Razdolnaya R. estuary at a distance of up to 28 km from the river mouth bar [41]. During this period, the water salinity is more than 34 PSU in

Amursky Bay and up to 26 PSU above the bar [41]. The average spring tide in Peter the Great Bay ranges from 15 to 20 cm, which makes it possible to classify the Razdolnaya R. estuary as a micro-tidal estuary with strong water stratification.

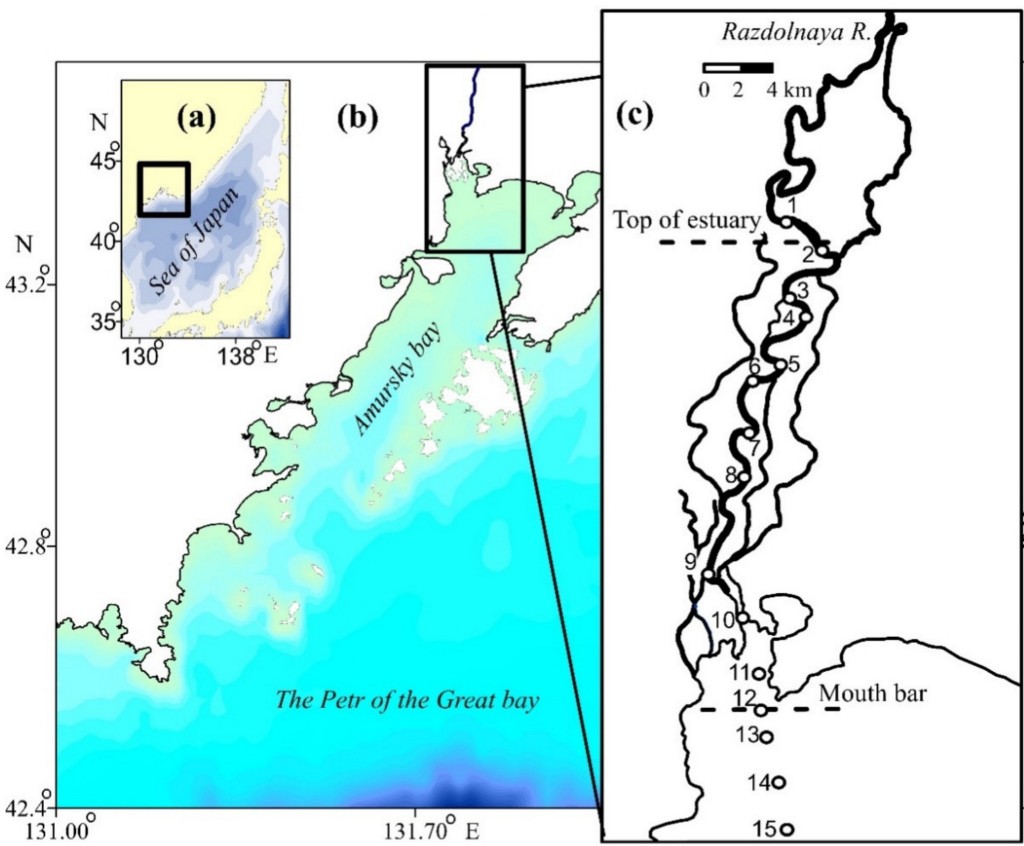

**Figure 1.** Map of study area: (**a**) Sea of Japan, (**b**) The Peter the Great Bay, (**c**) position of monitoring stations at Razdolnaya R. estuary.

## 2.2. Field Work, Hydrological Surveys and Water Sampling

We carried out a series of five surveys at 15 monitoring stations (St.) in Razdolnaya R. estuary (Figure 1) at roughly week-long intervals from 22 Jan. 2022 to 23 Feb. 2022. We probed the water with an RBR maestro multi-channel logger (RBR Ltd., Ottawa, ON, Canada) with an 8 Hz sampling rate. The following properties were logged: pressure, temperature, electrical conductivity, PAR, Chl-a fluorescence; chromophoric OM fluorescence, dissolved oxygen (DO) phosphorescence, and turbidity. The ice thickness was also recorded at the stations. The first four surveys took about five hours each. The fifth, comprehensive survey was taken when the ice thickness reached its highest value on Feb. 23. Surveys also included water sampling from the surface and near-bottom layers using 5 L Niskin bottles. Samples were analyzed at the onshore laboratory on the day of sampling to obtain the following parameters: salinity, pH, total alkalinity (TA), nutrients (silica, phosphorus, nitrates, nitrites, and ammonium), and Chl-a. The phytoplankton species composition was analyzed in the sample with the highest Chl-a concentration. We obtained the results of measurements of Chl-a fluorescence in Feb. of 2014 using a fluorescence sensor on the multiparametric Water Quality Monitor ((WQM) Wet-Labs, Philomath, OR, USA) installed on the bottom layer at St. 8. Additionally, we monitored the DO in Feb. 2016 at St. 5 using the multi-channel logger RINKO-Profiler with an optical fast DO sensor (JFE Advantech Co., Ltd., Nishinomiya, Japan).

### 2.3. Laboratory Analysis

The preparation of a water sample for the Chl-a determination was performed as follows. Firstly, 1 L water sample was filtered through 2 μm and extracted from the filters in a 90% acetone solution. The optical density of light absorption in the extracts was measured using a Shimadzu UV-3600 spectrophotometer (Shimadzu, Kyoto, Japan). Before measuring the pheophytin content, the extract was pre-acidified with 2–3 drops of a hydrochloric acid solution in acetone.

The ammonium concentration was determined using the indophenols method. Nitrates, nitrites, dissolved silicates (DSi), and dissolved inorganic phosphorus (DIP) were measured using standard colorimetric methods. The details of the methods used for the nutrient analyses are given in Grasshoff et al. [42]. The sum of the ammonium, nitrate, and nitrite concentrations was the dissolved inorganic nitrogen (DIN). The detection limit was 0.01 μmol/L for the phosphate and nitrite, and 0.02 μmol/L for silicate.

A potentiometric method was applied to determine pH. pH was measured at 10 °C using a cell without a liquid junction [43] and reported on the total hydrogen ion concentration scale [44]. The precision of pH measurements was about ±0.004 pH units. TA analysis was carried out through direct colorimetric titration with hydrochloric acid in an open cell according to Bruevich's method [43,45]. TA measurements were performed with a precision of ±3 μmol/kg. The $pCO_2$ and pH in situ were calculated from the measured pH and TA using a commonly known procedure [46]. The software used for statistical analyses was MS Excel 2019. The spatial distribution maps were developed using the program Surfer 9 (Golden Software).

## 3. Results

### 3.1. Hydrological Conditions

The observations were made during a steady state salt-wedge penetration to about 29 km from the river mouth bar (Figure 2). Further, salt-wedge penetration into the estuary was limited to the sandy channel between St. 1 and St. 2, with a depth of less than 0.5 m. As our observations showed, it was frozen to the bottom. There was a general decrease in the salinity of the bottom layer, which was most pronounced for St. 10, where we observed a decrease from 30 to 25 PSU (Figure 2). The freshening of the bottom layer was accompanied by a gradient decrease in the halocline layer, which can be distinguished by the salinity jump from 8 to 25 PSU (Figure 2).

The temperature of the sea and river waters was about −1.6 °C and 0 °C, respectively. The near-bottom layer temperature increased up to +2.3 °C during the observation period in the salt-wedge region (Figure 3).

The ice thickness on the transect as a whole increased during the observation period (Table 1). However, in the area of the bar on St.12, the thickness of the ice remained almost unchanged or even decreased in early Feb.

### 3.2. Chl-a Concentration

Figure 4 shows the intense phytoplankton bloom outbreak in a water layer below the halocline throughout the salt-wedge region. At the beginning of the observations (Jan. 22), the area of maximum concentration (40 μg/L) was located 5.5 km from the mouth bar at St. 10. During the next three surveys (Jan. 28, Feb. 4, Feb. 11), the maximum Chl-a concentration spread to the upstream estuary to the locations at 19.7 and 21.2 km from the mouth bar (St. 6 and 5, respectively), where Chl-a concentration increased from ~15 to 100 μg/L. After that, on Feb. 11 and Feb. 23, a decrease in the maximum Chl-a concentration was observed, which started from the area of St. 10, where the maximum Chl-a was initially observed on Jan. 22.

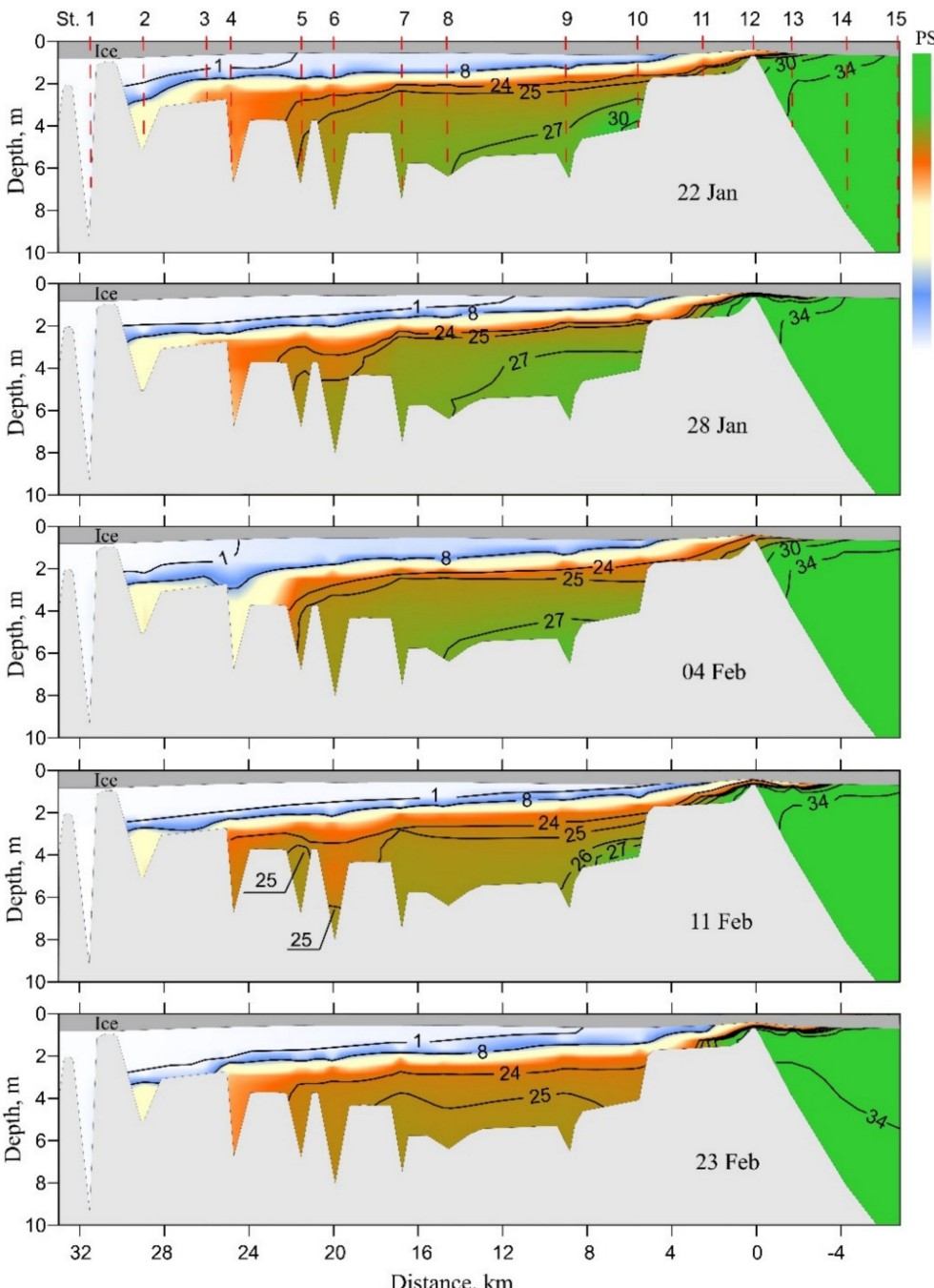

**Figure 2.** Salinity data on transect obtained from 22 Jan. 2022 to 23 Feb. 2022. The position of monitoring stations is shown by dotted red lines. Positive values on the *x*-axis—distance from the river mouth bar (St. 12) to upstream, negative values—to downstream.

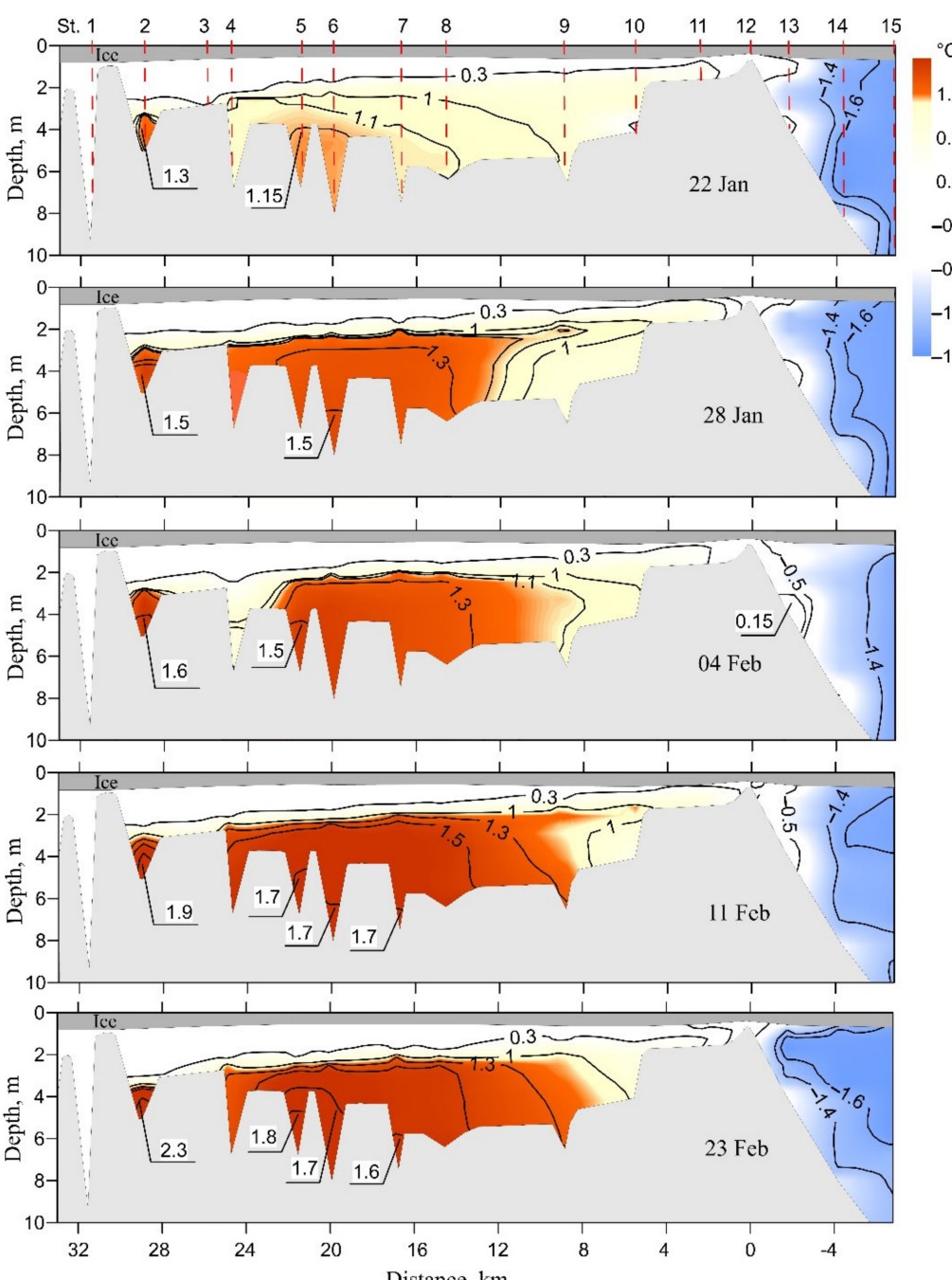

**Figure 3.** Water temperature on transect obtained from 22 Jan. 2022 to 23 Feb. 2022. The position of monitoring stations is shown by dotted red lines. Positive values on the *x*-axis—distance from the river mouth bar (St. 12) to upstream, negative values—to downstream.

**Table 1.** Ice thickness (cm).

| Data | \multicolumn{15}{c}{Station Numbers} | | | | | | | | | | | | | | |
| --- | --- | --- | --- | --- | --- | --- | --- | --- | --- | --- | --- | --- | --- | --- | --- |
| | 1 | 2 | 3 | 4 | 5 | 6 | 7 | 8 | 9 | 10 | 11 | 12 | 13 | 14 | 15 |
| 22-Jan. | 75 | 70 | 65 | 60 | 65 | 64 | 56 | 56 | 60 | 56 | 50 | 40 | 53 | 48 | |
| 28-Jan. | 79 | 73 | 66 | 64 | 65 | 65 | 58 | 56 | 61 | 58 | 51 | 42 | 56 | 51 | 45 |
| 4-Feb. | 83 | 75 | 68 | 65 | 67 | 67 | 56 | 57 | 62 | 57 | 50 | 35 | 50 | 55 | 49 |
| 11-Feb. | 81 | 76 | 70 | 69 | 70 | 70 | 56 | 55 | 62 | 58 | 50 | 40 | 64 | 60 | 54 |
| 23-Feb. | 90 | 79 | 66 | 67 | 69 | 69 | 55 | 55 | 61 | 59 | 52 | 41 | 70 | 66 | 60 |

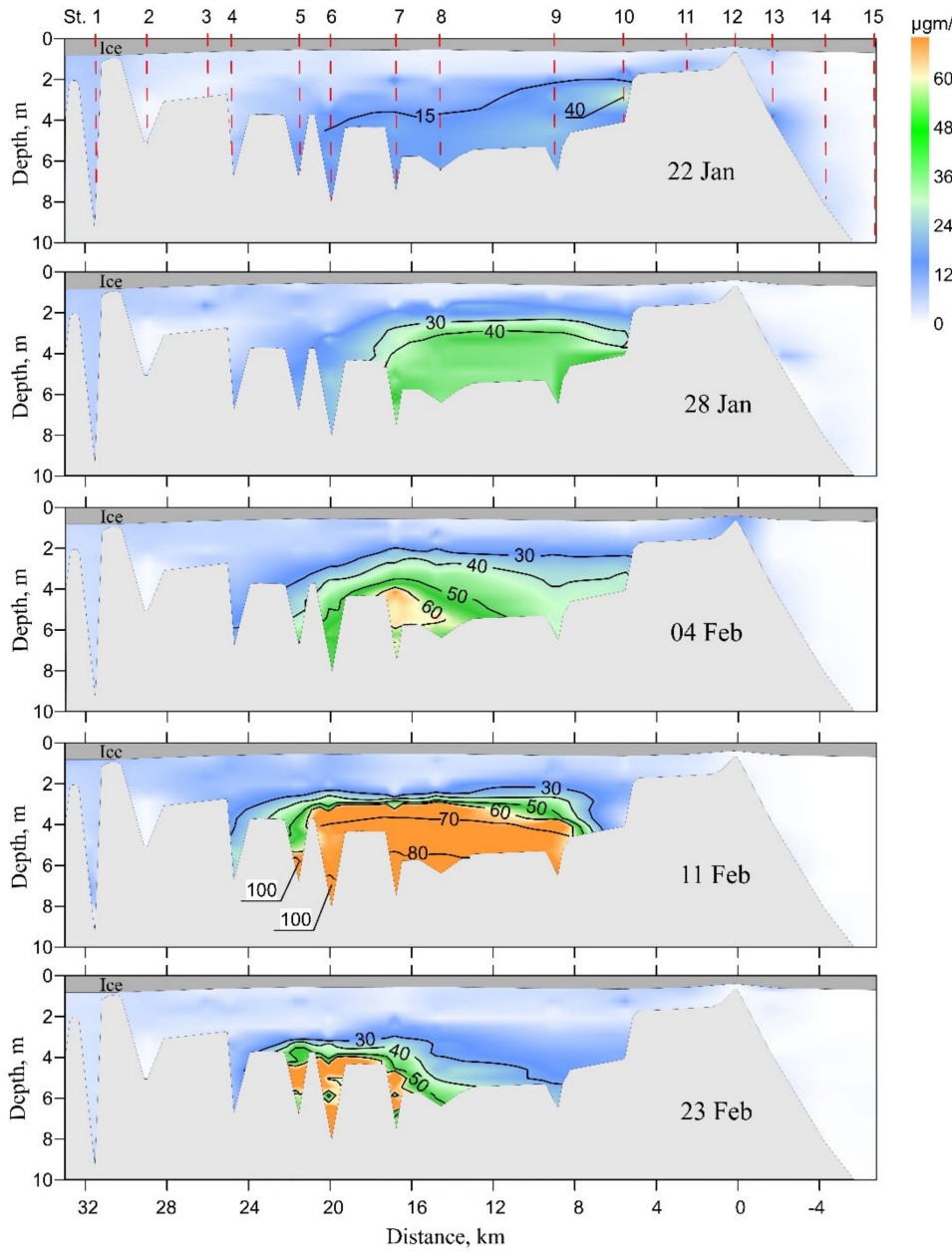

**Figure 4.** Chl-a concentration on transect obtained from 22 Jan. 2022 to 23 Feb. 2022. The position of monitoring stations is shown by dotted red lines. Positive values on the *x*-axis—distance from the river mouth bar (St. 12) to upstream, negative values—to downstream.

We obtained a relatively good convergence in the Chl-a concentrations for the two methods using St. 6 and 10 survey data (Table 2). The discrepancies for St. 4, 5, 7, 8, and

9 (Table 2) can be partially explained by the Chl-a concentration gradients and the depth difference of about 0.5 m between the probe and water sampler positions in the water layer.

**Table 2.** Chl-a and pheophytin concentrations obtained by profiling and spectrophotometry for the Feb. 23 survey samples. The "s" and "b" subscripts denote the surface and the bottom layer of water, respectively.

| | Profiling | Spectrophotometry | |
| --- | --- | --- | --- |
| St. | Chl-a, µg/L | Chl-a, µg/L | Pheo, µg/L |
| 1s | 7.8 | 5.6 | 0.7 |
| 2s | 9.4 | 6.0 | 1.3 |
| 2b | 1.0 | 1.6 | 1.9 |
| 4s | 6.1 | 5.8 | 2.2 |
| 4b | 19.1 | 8.5 | 2.2 |
| 5s | 4.2 | 12.8 | 1.6 |
| 5b | 60.0 | 48.9 | 2.6 |
| 6s | 6.8 | 11.5 | 1.7 |
| 6b | 70.0 | 74.2 | 2.7 |
| 7s | 3.3 | 10.3 | 1.4 |
| 7b | 60.0 | 32.7 | 2.5 |
| 8s | 8.0 | 12.8 | 3.5 |
| 8b | 50.0 | 36.9 | 3.5 |
| 9s | 6.8 | 11.3 | 1.2 |
| 9b | 43.7 | 33.0 | 2.2 |
| 10s | 10.2 | 20.8 | 2.9 |
| 10b | 20.0 | 17.6 | 1.3 |
| 11b | 5.5 | 20.9 | 2.1 |
| 12b | 1.0 | 17.1 | 2.5 |
| 13s | 0.8 | 7.0 | 0.5 |
| 13b | 0.2 | 3.2 | 0.5 |
| 14s | 0.2 | 0.2 | 1.7 |
| 14b | 0.1 | 1.3 | 0.3 |
| 15s | 0.3 | 1.1 | 0.4 |
| 15b | 1.2 | 5.1 | 0.3 |

According to WQM data, in 2014, a similar intensive increase in Chl-a concentration was observed in Feb. in the salt-wedge region (Figure 5). After reaching a maximum of about 45 µg/L in Feb., the concentration of Chl-a generally decreased in March.

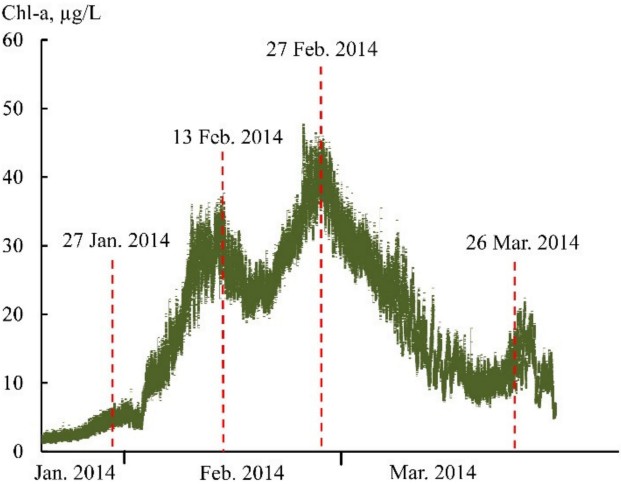

**Figure 5.** Long-term measurements by a bottom mooring probe WQM at St. 8 in 2014.

### 3.3. DO Concentration

The formation of the DO concentration maximum is generally attributed to the spreading of the Chl-a maximum across the salt-wedge region (Figure 6): (1) at the beginning of the observations (Jan. 22), the area of the maximum DO concentration (400 μmol/kg) was located 5.5 km from the estuary bar; (2) during the next three surveys (Jan. 28, Feb. 4, and Feb. 11), the DO concentration maximum spread up the estuary channel and increased to 567 μmol/kg (Figure 6) with the $O_2$ addition reaching 11 μmol/(kg day) for 20 days; (3) on 23 Feb., a decrease in the DO concentration was observed in the salt-wedge area.

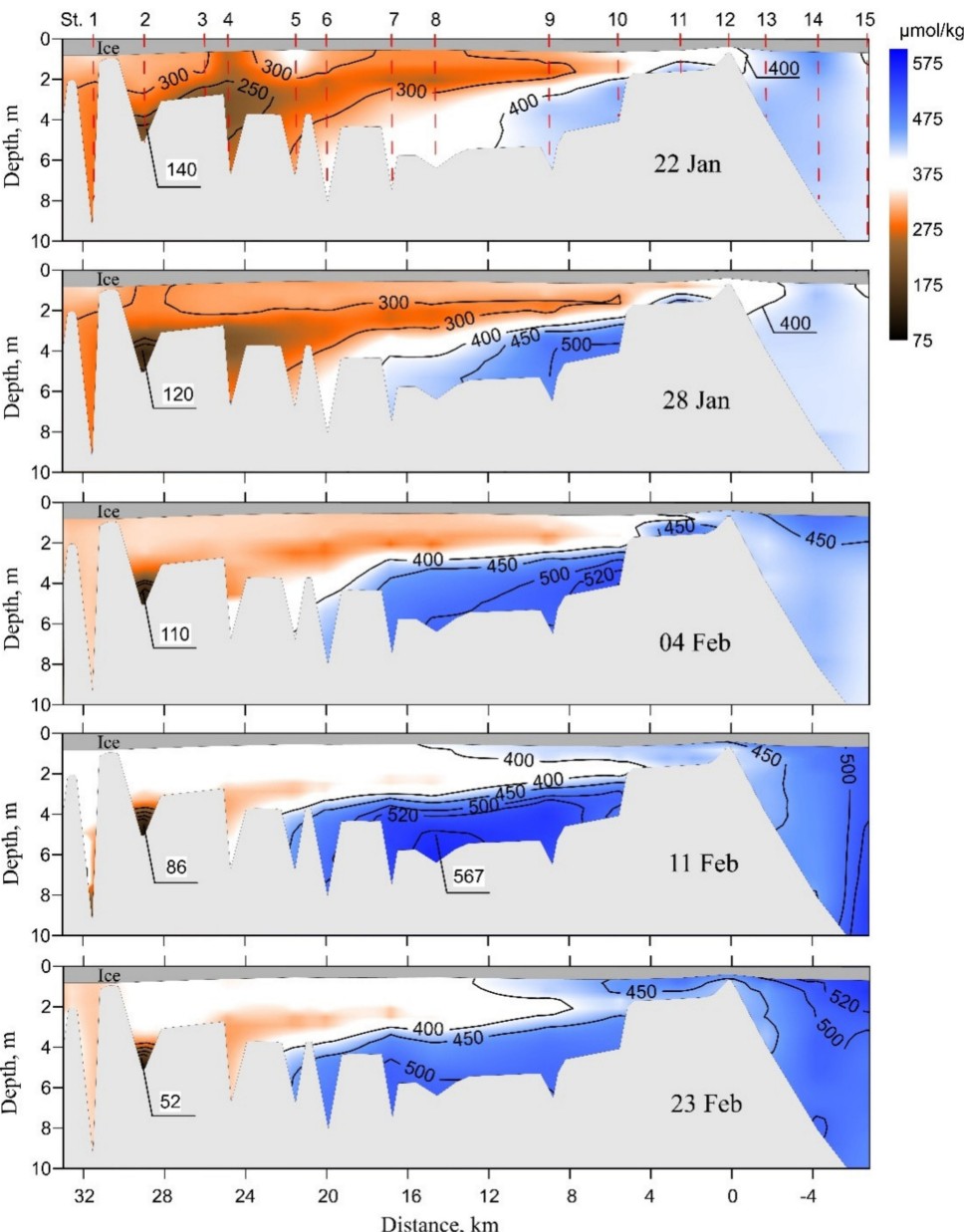

**Figure 6.** DO concentration on transect obtained from 22 Jan. 2022 to 23 Feb. 2022. The position of monitoring stations is shown by dotted red lines. Positive values on the *x*-axis—distance from the river mouth bar (St. 12) to upstream, negative values—to downstream.

The opposite temporal trend was observed with decreasing DO concentrations near the salt-wedge boundary at St. 2 (Figure 6). In this case, the DO concentration fell from 140 to 52 μmol/kg over the survey period, and was below the hypoxia threshold for the aquatic ecosystem, being in the range of 63–89 μmol/kg [6].

Figure 7 shows the fast-changing DO saturation from superoxia to hypoxia after the ice was covered with snow and the ability of PAR to penetrate through the ice was impeded.

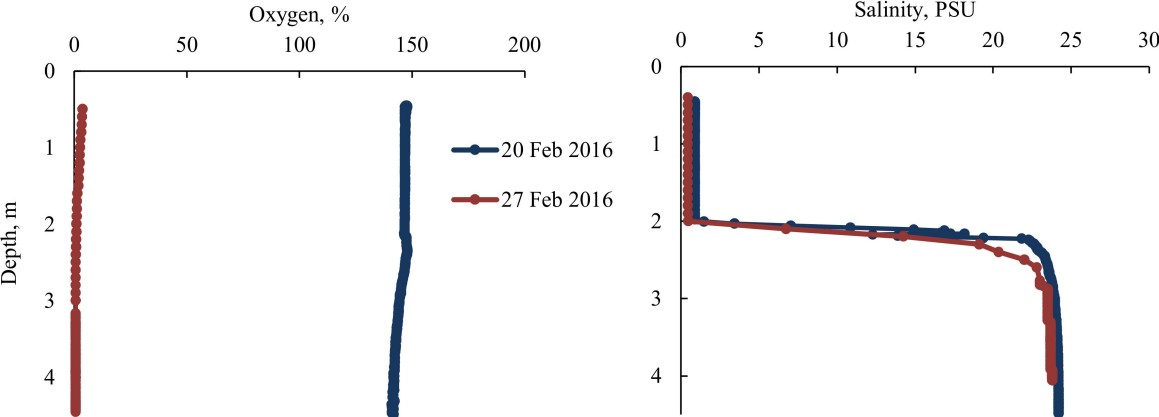

**Figure 7.** Vertical distribution of DO and salinity before (20 Feb. 2016) and after (27 Feb. 2016) snowfall at St. 5.

### 3.4. Chl-a Vs. AOU Curve

Apparent oxygen utilization (AOU) is a parameter for examining the production/destruction balance with regard to OM. Generally, the relationship between AOU and Chl-a concentration throughout the estuary has a negative slope (Figure 8). The two areas with the lowest AOU value (and the highest DO value) were identified in the salt-wedge area (203 µmol/kg) and the Amur Bay at St. 15 (167 µmol/kg). However, the maximum Chl-a concentration in the salt wedge was higher than that in the seawater by a factor of about 10 (Figure 8). Such a drastic difference between the Chl-a and AOU dependences for the two types of waters may be attributed to two factors: (a) a more intense level of OM destruction in the salt-wedge region, because in this case autochthonic and allochthonic components exist in OM; (b) there is more intensive grazing in seawater. The domination of OM destruction was most clearly manifested in the area of St. 2 where the AOU extremum was up to 322 µmol/kg.

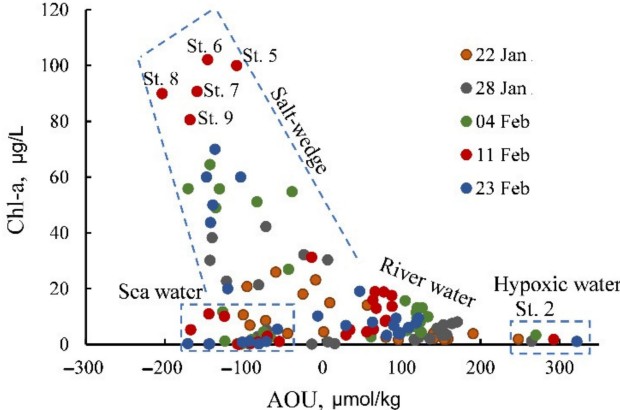

**Figure 8.** Chl-a vs. AOU curve.

Considering the above, the Chl-a and AOU relationship can take three different forms in three different types of waters: (1) waters dominated by OM destruction, with the maximum AOU value and minimum Chl-a concentration (up to 1 µg/L) at St. 2; (2) waters dominated by production with an increased DO concentration but a relatively low Chl-a concentration (up to 10 µg/L) beyond the salt-wedge; (3) and waters with intensive photosynthesis resulting from phytoplankton bloom under the halocline along the salt-

wedge at St. 5–10, with negative AOU values (the water is supersaturated compared to atmospheric oxygen) and the maximum Chl-a concentration (about 100 µg/L).

### 3.5. pH, pCO₂

As Figure 9 shows, by the end of the observation period $pCO_2$ decreased to 108 µatm and pH increased to 8.52 pH at a 25.0 PSU salinity (St. 7) in the layer of water below the halocline with the highest Chl-a concentration and the lowest AOU. The opposite pattern was observed outside the blooming area on St. 2, where the maximum AOU value was reached, the $pCO_2$ value was 4454 µatm, and the pH dropped to 7.0 at a salinity of 15.2 PSU.

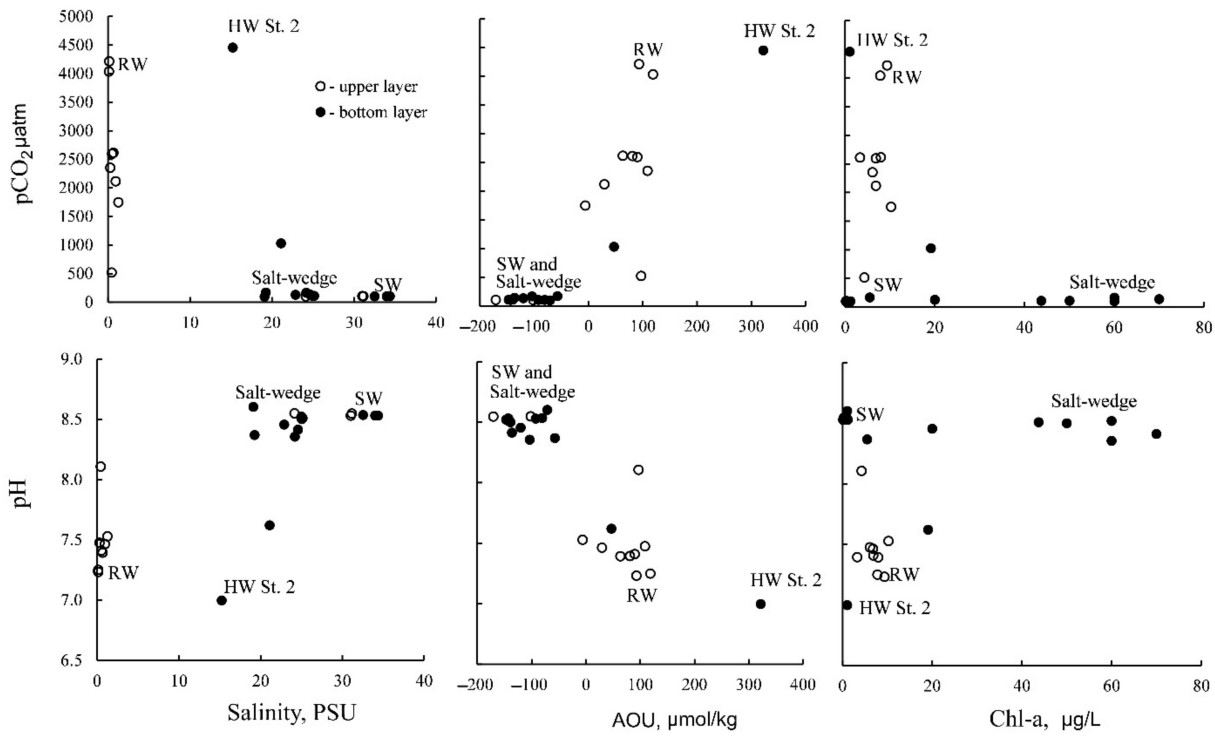

**Figure 9.** pH and $pCO_2$ vs. salinity, AOU, and Chl-a. RW: river water, SW: seawater, HW: hypoxic water.

## 4. Discussion

During the Feb. 11 survey, *Thalassiosira nordenskioeldii Cleve* was the predominant species of phytoplankton in the near-bottom layer in the area of the maximum Chl-a, St. 7. In Amur Bay, this species usually dominates at water temperatures from −1.8 to 0 °C and at 33.0–35.0 PSU salinity [40]. It was previously noted in other salt-wedge estuaries that marine diatoms (*Skeletonema*, *Thalassiosira*, and *Chaetoceros* species) dominated in the Chl-a maximum layer below the halocline [33,47,48]. The key points of discussion are the factors stimulating phytoplankton bloom outbreaks in a relatively short period and the possible implications for estuarine ecosystems.

### 4.1. Light Conditions

The estuary in the bloom area was covered with crystal-clear ice formed from the river water. The PAR intensity at the ice surface at noon reached 2000 µMol quanta/(m² s) (data from the probe before immersion into the water), and in the water, just under the ice, it was in the range of 100–250 µMol quanta/(m² s).

To analyze the vertical variability of PAR and the Chl-a concentration in under-ice water, we chose two stations: St. 7, the deepest station, which had high photosynthetic intensity under the halocline layer; and St. 5, with the maximum Chl-a concentration in the bottom layer of about 1 m thick (Figure 10). The light conditions under the halocline

over the observation period were generally unchanged (Figure 10), since differences in ice thickness do not significantly affect the PAR intensity in the under-ice water.

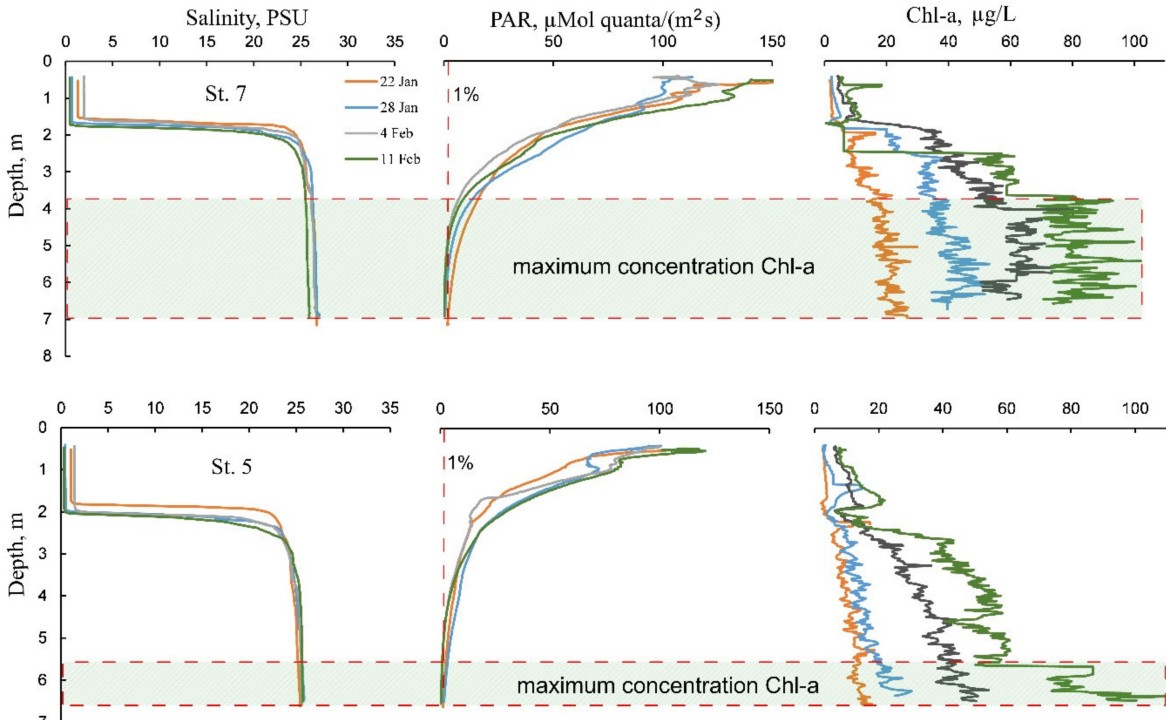

**Figure 10.** Vertical distribution of salinity, PAR, and Chl-a at St. 7 (**top panel**) and St. 5 (**bottom panel**) (note: survey 5 is not shown). The red vertical dashed line indicates the threshold (1% of the PAR intensity) relative to the surface water layer.

The penetration of light is largely governed by the depth of the snow on the ice [49]. We consider as a special case the situation in Feb. 2016 at St. 5 after a snowfall. PAR did not penetrate under the ice when the thickness of the snow on the ice was about 20 cm, and DO decreased from 150% saturation (500 μmol/kg) to about 1% in a week (Figure 7).

### 4.2. The Concentration of Nutrients

During the winter, the main flux of nutrients into the Razdolnaya R. estuary comes from municipal wastewater [36]. The volume of these effluents generally does not change during winter or even throughout the year.

The DSi/DIN molar ratio was above one everywhere, indicating that the DSi concentration was sufficient for diatom algae growth [50]. At the end of the observation period, the low DIP concentration may have been a factor limiting further bloom, since the DIN/DIP ratio reached 236 near the salt-wedge (Figure 11), i.e., significantly exceeding the Redfield DIN/DIP ratio = 16 [51]. The exception was the bottom seawater at the most remote stations beyond the mixing zone (St. 14 and 15) where the ratio was below 10. The previous studies on the estuary [35,41,52] also indicated a relatively high DIN/DIP ratio. To some extent, this is due to the depletion of the phosphates as they are bound by iron [53]. The content of dissolved and suspended iron in the Razdolnaya R. is significantly higher than in marine waters [54]. Against the background of photosynthesis intensification, we observed an almost complete DIP removal and, accordingly, a relatively high DIN/DIP ratio.

In the salt-wedge region outside the bloom region at St. 2, the dominance of OM destruction was detected as the extremum of ammonium (Figure 11) with the decreasing DO (Figure 6) and increasing $pCO_2$ (Figure 9). OM destruction with DO, in accordance with Redfield stoichiometry [51], is accompanied by an increase in nutrient concentrations

in water, which was observed in some parts of the estuary at salinity ~19 in shallow waters
(St. 11 and 12).

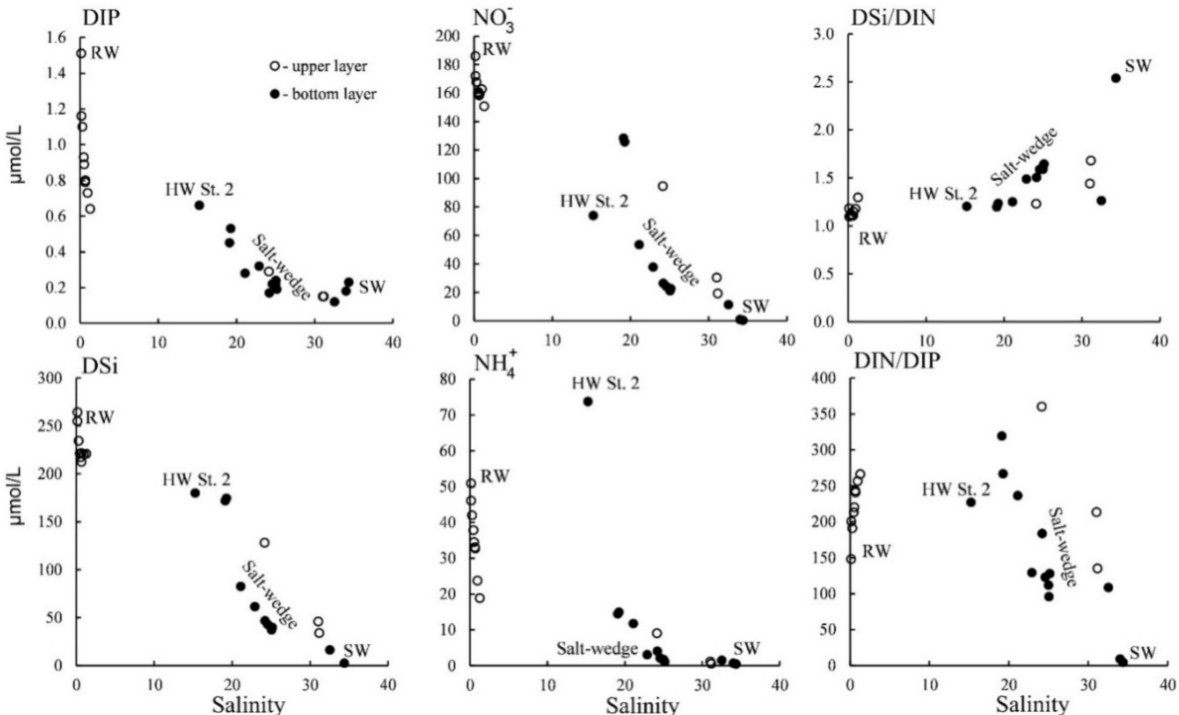

**Figure 11.** Nutrient concentration vs. salinity in near-bottom and surface water layers. 23 Feb. 2022
RW: river water, SW: seawater, HW: hypoxia water with low oxygen concentration.

### 4.3. Water Layer Stability

The ice thickness during the study period was comparable to the depth of the estuary
bar (~50 cm), i.e., it was frozen almost to the bottom (Table 1). In this case, the water
exchange rate between the estuary channel and the receiving basin decreases. The river
discharge during the ice formation period is minimized, and the estuary in this case can
be compared to a closed lake ecosystem to a certain extent. In many ice-covered lakes,
there is a massive growth of diatom algae in winter due to the water layer stability under
constant light conditions [55,56]. We hypothesize that *Thalassiosira nordenskioeldii* cells were
brought upstream during the period when the river flow decreased and the salt-wedge
intrusion regime was established, and subsequently, that the water layer stability was the
key reason for the phytoplankton bloom outbreaks. This hypothesis is confirmed by the
fact that the initial moment of the bloom generally coincided with the period when ice
thickness in the mouth bar at St. 12 reached the maximum value. Outside the salt-wedge
area, the ice thickness increased over the entire observation period (Table 1). Since the
same moment, there was a salinity decrease and a bottom layer temperature increase. The
resulting combination indicates a limited inflow of relatively cold sea water during the
observation period (Figures 2 and 3).

However, the phytoplankton bloom during the ice formation in the Razdolnaya R.
estuary does not occur annually. We also observed a bloom during the harsh winter in
February 2014 (Figure 5), as the river ice thickness reached 89 cm. No phytoplankton
blooms were observed during the relatively mild 2008 and 2020 winters with smaller ice
cover [41,52]. The ~90 cm river ice thickness (Table 1) indicated a relatively cold winter. In
such winters, a distinct positive temperature anomaly in the bottom layer in the salt-wedge
region occurs in February, e.g., in February 2014 [57] and in February 2022 (current results).
This indicates a decrease in water exchange over the river mouth bar in cold years.

*4.4. Effects of the Bloom on the Estuarine Ecosystem*

This study considers a bloom at relatively high PAR intensity in under-ice water with no snow on the ice. The second factor contributing to the high PAR intensity is the low concentration of suspended solids during the ice formation period: ~5 mg/L [36]. The suspended sediment concentrations in the Razdolnaya R. can exceed 2800 mg/L in summer and fall during the typhoon season. Therefore, during the ice-free period, the PAR intensity decreases rapidly as the depth increases, and the bloom occurs exclusively in the surface water layer and is accompanied by the hypoxia of bottom waters in the salt-wedge area [35]. The second option for bloom in the ice-free period occurs during a high-water period, with the immediate eutrophication of the Amur Bay, which is the outer part of the Razdolnaya R. estuary, and the subsequent hypoxia of near-bottom waters [10]. The carbonate system parameters (pH and $pCO_2$) are connected through some thermodynamic relationships and reflect the direction of the OM production/destruction balance [46]. This study demonstrates a third possible bloom option with a Chl-a extremum in the bottom layer under the halocline with an $O_2$ and pH increase, and $CO_2$ removal. In some years, when snowfalls occur and the ice is covered with snow, the intensity of PAR under the ice decreases and photosynthesis stops. As a result, oxygen levels decrease quickly due to the destruction of diatom biomass and conditions become anaerobic (Figure 7). The decrease in oxygen levels leads to the death of small fish, which freeze to the lower boundary of the ice, which we observed visually. Subsequently, the snow is quickly swept away by the wind, and in this case, an increase in DO can be expected again, since the light conditions become sufficient for photosynthesis.

The vertical distribution of the Chl-a concentration has two features: the maximum Chl-a is spread over the entire water layer under the halocline (St. 7), and at St. 5 it was accumulated near the bottom (Figure 10). The deepening of the Chl-a extremum may be attributed to the aggregation of "sea snow" [58]. This phenomenon is a result of living phytoplankton cell gluing by polysaccharides [59]. Aggregation may be delayed because large aggregates with neutral buoyancy or even slow upfloat rates associated with the formation of gas bubbles inside the flakes often persist for several days [60]. The delayed aggregation can partially explain the existence of a Chl-a maximum in the entire water layer below the halocline in the most intense photosynthesis area at St 7. The two stations considered (St. 5 and St. 7) show that the aggregation processes will manifest differently throughout the salt-wedge region. The presence of more intensive aggregation in stretches is further indicated by the highest concentration of dissolved OM in the pore water of the sediment in these stretches, which can be up to 40 mgC/L [61]. During the studies of the bottom sediments at St. 5, we found some accumulations of polychaetes *Tylorrhynchus heterochetus* with a high density of up to 3000 organisms/m². This indicates a sufficient amount of nutrients for the hydrobionts in this stretch.

Usually, the common organisms consuming diatoms are Copepoda, which in turn are fish food [62]. According to ichthyological observations in the Razdolnaya R. estuary in winter, the fish biomass in the salt-wedge area ranges from 71 to 374 g/m², while in summer it does not exceed 10 g/m² [63]. It is likely that the growth of zooplankton lagged behind the diatom growth by about two weeks, and the feeding of anadromous fish species resulted in a photosynthesis rate decrease due to phytoplankton being eaten by zooplankton after the bloom peak in Feb. (Figures 4, 5 and 8).

## 5. Conclusions

The weakening of the water exchange during the ice formation period is a key factor that activates the phytoplankton bloom outbreak under favorable light conditions and high nutrient concentrations. This study highlights the fundamental role of water exchange in the river estuary ecosystems with cultured watersheds. The waters with the maximum concentration of Chl-a were dominated by marine microalgae that emerged from the sea to the bottom layer under the halocline during the stable low-water period in winter. The under-ice blooms were accompanied by the supersaturation of bottom waters with oxygen

and undersaturation with carbon dioxide, as opposed to blooms during free channel periods which occur in the surface water layer with bottom-water hypoxia. Under-ice phytoplankton blooms in the eutrophic estuary may be accompanied by the formation of hypoxia for a period of about one week after a snowfall, when the ability of PAR to penetrate through the ice is impeded by snow cover. The management of nutrient fluxes and a better understanding of the production/destruction balance under the control of natural and human factors may be vital in mitigating algal blooms in eutrophic estuaries, such as the Razdolnaya R. estuary.

**Author Contributions:** P.S.: Writing—Original draft, Conceptualization, Methodology, Formal analysis, Investigation, Funding acquisition. P.T. (Pavel Tishchenko): Writing—Review and Editing, Investigation, Conceptualization, Supervision, Funding acquisition. G.P.: Writing—Review and Editing, Investigation, Formal analysis. Y.B.: Methodology, Formal analysis, Investigation. P.T. (Petr Tishchenko): Writing—Review and Editing, Investigation, Formal analysis. M.S.: Formal analysis, Investigation. E.S.: Formal analysis, Investigation. Y.F.: Formal analysis, Investigation. All authors have read and agreed to the published version of the manuscript.

**Funding:** This work was supported by the Russian Science Foundation (Project No. 21-77-00028) at Ilichev Pacific Oceanological Institute, Far Eastern Branch Russian Academy of Sciences (Reg. No. 121-21500052-9, AAAA-A20-120011090005-7).

**Institutional Review Board Statement:** Not applicable.

**Informed Consent Statement:** Informed consent was obtained from all subjects involved in the study.

**Data Availability Statement:** Not applicable.

**Conflicts of Interest:** The authors declare no conflict of interest.

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
