# Peer review of "O2 and CO2 Responses of the Synaptic Period to Under-Ice Phytoplankton Bloom in the Eutrophic Razdolnaya River Estuary of Amur Bay, the Sea of Japan"

_jmse, doi:10.3390/jmse10121798_

Round 1

Reviewer 1 Report (Previous Reviewer 2)

The details can be seen in attachment

Author Response

We appreciate for your precious time in reviewing our paper and providing valuable comments. All modifications in the manuscript have been highlighted in file “jmse-2037159 11 Nov 2022”.

Reviewer 2 Report (New Reviewer)

The article contains interesting factual material and can be published in the journal after some improvements:

1. Figure 7-11 and the corresponding description can be found in the Discussion section. But this is the actual material and it should be moved to the Results. This will improve the perception of the article.

2. Check the spelling in the Reference. There are errors, for example 20. Sakshaug, E

Author Response

We appreciate for your precious time in reviewing our paper and providing valuable comments. All modifications in the manuscript have been highlighted in file “jmse-2037159 11 Nov 2022”.

This manuscript is a resubmission of an earlier submission. The following is a list of the peer review reports and author responses from that submission.

Round 1

Reviewer 1 Report

Adjustments to the inclusion of figures in the discussion section are recommended, as well as paragraphs in the results section that correspond to the discussion.
